# In Silico Analysis of Pacific Oyster (*Crassostrea gigas*) Transcriptome over Developmental Stages Reveals Candidate Genes for Larval Settlement

**DOI:** 10.3390/ijms20010197

**Published:** 2019-01-08

**Authors:** Valentin Foulon, Pierre Boudry, Sébastien Artigaud, Fabienne Guérard, Claire Hellio

**Affiliations:** 1Laboratoire des Sciences de l’Environnement Marin (LEMAR), UMR 6539 CNRS/UBO/IRD/Ifremer, Institut Universitaire Européen de la Mer, Technopole Brest-Iroise, Rue Dumont d’Urville, 29280 Plouzané, France; sebastien.artigaud@univ-brest.fr (S.A.); fabienne.guerard@univ-brest.fr (F.G.); claire.hellio@univ-brest.fr (C.H.); 2Ifremer, Laboratoire des Sciences de l’Environnement Marin (LEMAR), UMR 6539 CNRS/UBO/IRD/Ifremer, Centre Bretagne, 29280 Plouzané, France; pierre.boudry@ifremer.fr

**Keywords:** *Crassostrea gigas*, Pacific oyster, pediveliger larvae, bioadhesive, transcriptome

## Abstract

Following their planktonic phase, the larvae of benthic marine organisms must locate a suitable habitat to settle and metamorphose. For oysters, larval adhesion occurs at the pediveliger stage with the secretion of a proteinaceous bioadhesive produced by the foot, a specialized and ephemeral organ. Oyster bioadhesive is highly resistant to proteomic extraction and is only produced in very low quantities, which explains why it has been very little examined in larvae to date. In silico analysis of nucleic acid databases could help to identify genes of interest implicated in settlement. In this work, the publicly available transcriptome of Pacific oyster *Crassostrea gigas* over its developmental stages was mined to select genes highly expressed at the pediveliger stage. Our analysis revealed 59 sequences potentially implicated in adhesion of *C. gigas* larvae. Some related proteins contain conserved domains already described in other bioadhesives. We propose a hypothetic composition of *C. gigas* bioadhesive in which the protein constituent is probably composed of collagen and the von Willebrand Factor domain could play a role in adhesive cohesion. Genes coding for enzymes implicated in DOPA chemistry were also detected, indicating that this modification is also potentially present in the adhesive of pediveliger larvae.

## 1. Introduction

The majority of bioadhesives secreted by animals are composed of proteins that allow permanent or reversible links to the substrate [1]. In the marine environment, these glues are efficient in wet conditions and could thus potentially represent useful alternatives to synthetic adhesives [2], particularly for biomedical applications [3]. The molecular composition of marine bioadhesives, especially proteins, can sometimes be difficult to characterize, however, due to their high resistance and small quantities, particularly for bioadhesives secreted at the larval stage. This issue has already been reported for bivalve mollusk larvae [4,5,6,7,8].

The Pacific oyster *Crassostrea gigas* (Thunberg 1973) is a benthic mollusck of the bivalve family with a two-phase life cycle. Its pelagic larvae adhere to a surface prior to metamorphosis. Larval settlement occurs at the pediveliger stage by secretion of a bioadhesive [4]. Overall molecular characterization of the adhesive secreted by the pediveliger larvae of *C. gigas* revealed its proteinaceous nature [4] and corroborate previous results published on pediveliger larval adhesive in other species [7,8,9,10]. However, the constitutive protein sequences of adhesive from *C. gigas* larvae remain unknown. The identification of genes involved in adhesion could be a useful first step towards protein identification that would enable us to successfully characterize the composition of *C. gigas* larval adhesive.

Numerous transcriptomic studies have recently been carried out on bioadhesive secretory organs. Rodrigues et al. (2016) used transcriptomics and proteomics approaches in cnidarians of the genus *Hydra*, to successfully pinpoint genes, proteins and enzymes potentially involved in adhesive composition and polymerisation [11]. A similar approach was used on the foot and byssus of *Chlamys farreri*, making it possible to understand scallop attachment [12]. Moreover, the transcriptome of the *Mytilus coruscus* foot allowed the identification of sequences with a strong homology to the adhesive sequences of other *Mytilidae* [13]. A transcriptomic study on adhesive glands of polychaetes of the *Sabellariidae* family recently described the phylogenetic evolution of certain adhesion genes and highlighted the importance of post-translational changes in adhesive proteins [14]. Transcriptomic analyses are described as an innovative and effective tool for determining candidate genes in marine organisms, but require validation by other molecular and functional investigations [1,15]. In *C. gigas* pediveliger larvae, the transcriptome of the adhesive gland is difficult to obtain due to the small size of the organism and the complexity of this organ.

However, the development of high-throughput nucleic acid sequencing methods (DNA and RNA) has led to a significant increase in the number of sequences available in generalist or specific databases (for the transcriptome of *C. gigas*: Riviere et al. (2015) [16]). The use of appropriate informatics tools makes it possible to identify sequences of interest in databases, compare them, analyze them and define their potential biological roles [17]. Many studies, known as in silico studies, use the available genomic data to identify genes involved in a defined biological process in order to answer to a working hypothesis. Sequence selection criteria (genomic expression rate, specificity of organs or certain stages of development, functional annotations) are defined according to the biological questions raised. Meta-analyses based on the exploration of published genomes are becoming increasingly common [18,19,20]. For example, in *C. gigas*, a recent study focusing on adult photosensitivity used this method to identify genes involved in this process [21]. Here, we propose to use a similar method to investigate bioadhesion of pediveliger larvae.

In *C. gigas*, adhesive is synthesized and stored before secretion from glands located in the foot [4]. Morphogenesis of the foot is a rapid process (24 to 48 h), specific to the pediveliger stage. This organ has locomotory, sensory, and secretory roles during the adhesion phase and disappears during metamorphosis (just a few hours after settlement). This indicates that adhesive synthesis is also a rapid process, resulting from significant and episodic cellular activity in the foot.

The presence of an mRNA in an organism at a given time is an indication of protein synthesis. The translation time of any given mRNA into protein is highly variable, however, from a few minutes to a few hours [22,23], and detection of an mRNA is not proof of the presence of the corresponding protein at a given time. However, identifying the genes expressed at a given time can still provide arguments for discussion about its involvement in a biological process. In our study, the first version of the genome assembly of the Pacific oyster *C. gigas*, published in 2012 by Zhang et al., is a particularly interesting resource [24]. The transcriptomic data also published in this article is available in Supplementary Table S14 “Transcriptomic representation of genes (RPKM) at different developmental stages and in different adult organs”. This dataset groups the RPKMs (read per kilobase million) of each gene at each stage of development, and in different tissue of adult oysters. These quantitative data make it possible to visualize the expression rate of each genomic sequence. Nine developmental stages defined by 38 sampling time from hours to days after fertilization and 11 different adult organs were analyzed in this article. The pediveliger stages were named P1 and P2 and correspond to larvae of 18 days old larvae (precisely sampled at 18 days and 45 min and at 18 days, 4 h and 35 min after fertilization).

The objective of this study was to identify genes in the transcriptomic data published with the genome of *C. gigas* that could have a potential role in the adhesion of the pediveliger larvae. The identification of these genes could allow us to suggest the probable protein composition of the adhesive and to pinpoint the biosynthesis pathways and molecular cascades involved in their secretion and cross-linking. The sequences specifically expressed at the pediveliger stage and the potential role of the corresponding proteins are presented. After functional annotation of the sequences, those of them with interesting adhesion characteristics can be considered as relevant candidates for future molecular investigations.

## 2. Results

Fifty-nine sequences were selected as being specifically expressed at the pediveliger stage of *C. gigas* (Table 1) according to the following selection criteria: RPKM [pre-pediveliger stage (LU1 and LU2)]/RPKM [pediveliger stage] > 0.7 * RPKM [pediveliger stage] and RPKM [other stages]/RPKM [pediveliger stage] > 0.2. This selection represents 0.23% of the 27,902 sequences from the Table S14 of Zhang et al. (2012) [24]. sequences had at least one predicted conserved domain and/or one repeat sequence based on analysis with InterPro [25] (Figure 1). Forty-two sequences had extracellular localization according to DeepLoc 1.0 [26]. Twenty-one sequences, or 35.6% of the selected sequences, were annotated as hypothetical proteins, indicating the absence of known functions from the databases. The number of uncharacterized sequences is slightly lower than the 41.8% of sequences annotated as hypothetical proteins in the database used as a whole.

## 3. Discussion

### 3.1. Sequences Involved in Reduction-Oxidation Reactions (Redox)

Among the fifty-nine selected sequences, three tyrosinase-like proteins (CGI_10009044, CGI_10014286 and CGI_10006802) and one peroxidase-like protein (CGI_10016593) are enzymes involved in reduction-oxidation (Redox) mechanisms. Tyrosinase is an oxidoreductase, also known as phenoloxidase, which allows hydroxylation (addition of an -OH group) of the aromatic part of tyrosine residues. Hydroxylated tyrosine, known as 3,4-dihydroxyphenylalanine (DOPA), can in turn be oxidized by tyrosinase, thus becoming a dopaquinone carrying two =O groups. DOPA-based marine bioadhesives are known to be sensitive to pH variations and to involve coacervation mechanisms [27,28]. Coacervation is a physicochemical mechanism allowing the spontaneous dissociation into two phases of a mixture of molecules due to their opposite charges. The most studied adhesive mechanism is mussel byssus, composed of a filament and a byssal plaque in contact with the substrate. In adult mussels, during the formation of the byssal plaque by the foot, the adhesive is secreted in a coacervated form, and polymerized by redox mechanisms mediated by the pH of the environment [27,29]. This strategy has also been described in the adhesive of the polychaetes *Sabellaria alveolata* and *Phragmatopoma californica* [28,30,31,32]. After secretion, the described DOPA-based adhesives combined with a coacervation mechanism had a foamy structure. However, the adhesive secreted by *C. gigas* larvae was described as a fibrous structure [4]. Phenoloxidase granules were reported in the main gland of the foot of pediveliger larvae of *O. edulis* by histochemistry [33]. The presence of phenoloxidase granules has not been confirmed in *C. gigas*, although secretion of byssal-like filaments by pediveliger larvae was observed before permanent adhesion at the end of the crawling phase [4]. It is possible that similar byssal secretion strategies could be used by pediveliger oyster larvae and adult mussels. Tyrosinase EKC29117 (CGI_10006802) was predicted for extracellular localization and presented a similarity of 47.95% (*E*-value: 6.5 × 10^−75^) to a byssal protein sequence from *Mytilus corsuscus* (ANN45959 | Byssal tyrosinase-like protein 2). The sequence EKC25254 (CGI_10014286) had 46.19% homology (*E*-value: 1.4 × 10^−84^) with an analogue protein (AKI87982 | Byssal tyrosinase-like protein-1) [34]. Interestingly, this last sequence had a C-type lectin domain (position 67–187), indicating probable linking to a polysaccharide. The C-type lectin domain could allow the immobilisation of the enzyme in the adhesive after secretion by linkage to a polysaccharide. This binding site could also act as an activation/inactivation site of the enzyme, as observed in tyrosinases implicated in melanin synthesis [35]. The presence of glycosylated active enzymes has also been reported in the adhesive of *Trichopterae* larvae [36].

Another sequence implicated in redox mechanisms, CGI_10016593 | EKC32997 | peroxidase-like protein, was also detected. This sequence had 44.8% similarity (*E*-value: 2.6 × 10^−57^) to Byssal peroxidase-like protein 1 of *Mytilus coruscus* [34]. Byssal peroxidases have also been detected in the foot of *Limnoperna fortunei* [37] and byssus of *Pinctada fucata* [38]. The role of these enzymes remains uncertain, but byssal peroxidases could be involved in the protection of byssus from oxidizing environments and degradation brought about by microorganisms. The sequence EKC32997 (CGI_10016593) had a signal peptide indicating the extracellular secretion of this protein. These enzymes could also be directly involved in cross-linking of the adhesive, as described in the larvae of *Hysperophylax occidentalis*, allowing the establishment of di-tyrosine bonds that stabilize adhesive fibers [36]. The presence of peroxidases was also reported in the basal disc of the polyp *Hydra magnipapillata* [11], at the interface of the adhesive plaque in the barnacle *Amphibalanus amphitrite* [39,40], and in the parathorax of the polychaetes *Sabellaria alveolata* and *Phragmatopoma caudata* [14]. In algae, haloperoxidases are also involved during adhesion, catalyzing the redox reactions of phenolic compounds in adhesive mucilages [41,42]. All these observations indicate an important role of this enzyme, particularly in the biosynthesis of DOPA and its derivatives.

The presence of tyrosinase and peroxidase coding sequences in our *C. gigas* pediveliger larva sequence selection could indicate a use of redox mechanisms in its adhesive with the presence of DOPA or phenol groups.

Two other sequences coding for enzymes involved in redox processes were selected: CGI_10010889 and CGI_10011324, annotated as carbonic anhydrase. These enzymes are closely related to calcification, are important regulators of the acid–base balance, and could be involved in the cellular regulation of CO_2_ at different cellular levels [43,44]. In molluscks, carbonic anhydrase activity is associated with shell calcification and active domains have been reported in nacrein protein [45]. This enzyme is strongly expressed in the mollusck mantle [46]. To date, carbonic anhydrase has never been reported in bioadhesive studies. It is therefore very likely that these two mRNA are expressed at the pediveliger stage in order to prepare the rapid calcification observed after metamorphosis. Also, it should be noted that the adhesion of pediveliger larvae in bivalve molluscks involves the shell as the upper interface. This is unique in mollusck adhesives since, for adult bivalves secreting byssal secretions, the binding between the adhesive and the body is provided by the tissues of the foot [27,47]. The binding between the shell and the adhesive in the pediveliger larvae of *C. gigas* could be strengthened by the action of carbonic anhydrase mobilizing carbonate from the shell. This selected set of genes specifically expressed at the pediveliger stage, encoding proteins involved in redox mechanisms, could therefore play a major role in the adhesion of *C. gigas*.

### 3.2. Proteases and Enzyme Inhibitors

Proteases were also detected in our selection. The sequence CGI_10007190 (EKC19955 | metalloproteinase) contained two Epidermal Growth-Factor (EGF) domains (positions 277–312 and 438–476) and a complement domain C1r/C1s, Uegf, Bmp1 (CUB) (position 320–438). The sequence EKC19956 has the same architecture as two EGF domains (positions 447–485 and 654–684), one CUB domain (position 329–447) and a meprin, an A-5 protein, and a receptor protein-tyrosine phosphatase mu domain (MAM) at the end of the sequence (position 920–1074). The sequence CGI_10020760 was annotated as Zinc metalloproteinase nas-15 (EKC31184), with a ZnMc domain (position 110–239). All these proteases were extracellular metalloproteinases which could play a role in the remodeling of the extracellular matrix during metamorphosis. Metalloproteinases have also been reported to play a role in synaptic systems and neural development. The larval transition to the pediveliger stage is accompanied by the development of the nervous system of the foot, which is largely innervated. It is probable that the presence of these sequences is related to this phenomenon. In addition, immediately after adhesion, the foot disappears during metamorphosis [48]; this tissue remodeling probably involves protease action.

Protease inhibitor sequences were also selected. The sequence CGI_10005578 (EKC25384 | hypothetical protein) was annotated with Gene Ontology indicating a metalloendoproteinase inhibitor molecular function. Recently, a protein with a similar function was identified in the foot and byssus of *Chlamys farreri* [12,49]. In this species, the protein Sbp8-1, which was described as an atypical metalloproteinase inhibitor, is a component of the byssus that is probably involved in the binding between the different byssal proteins. Two sequences annotated as serine protease inhibitors (CGI_10010154 and CGI_10010155) were specific to the pediveliger stage. The three sequences CGI_10010553, CGI_10010554 and CGI_10010557 had a trypsin inhibitor domain, and the sequences CGI_10010556 and CGI_10010557 had a serine protease inhibitor domain. Gene expression of serine protease inhibitor has been detected in the foot of *Mytilisepta virgata* [50] and *Chlamys farreri* [12].

In *C. gigas*, enzyme inhibitors could be involved in protecting the adhesive from degradation after secretion. Indeed, the presence of the adhesive secreted at the pediveliger stage (14 days post-fertilization) was still observable 72 days post-fertilization, indicating the robustness of this biomaterial.

### 3.3. Sequences Related to the Extracellular Matrix

Two hyaluronidase-related sequences were selected as specific to the pediveliger stage: CGI_10003099 and CGI_10003100. The role of these sequences was difficult to determine. Hyaluronidases had hydrolytic activity on hyaluronic acid and some forms of chondroitin sulfate, and could allow the remodeling of the extracellular matrix.

Six of the selected sequences were annotated as Fc receptors (IgGFc-binding protein): CGI_10010553, CGI_10010554, CGI_10010555, CGI_10010556, CGI_10010557 and CGI_10023170. These proteins are generally associated with immunity as they bind antigens from pathogens. In humans, some IgGFc-binding proteins are associated with mucus composition [51]. This type of protein has been identified in the mucus of *Crassostrea virginica* [52], where it was involved in the structure of the mucus via interactions with mucins. Mucus could be secreted as reversible adhesive by pediveliger larvae of *C. gigas* during the crawling phase. Indeed, contents of foot glands A and B, implicated in crawling, could potentially be related to mucus [4].

It is probable that IgGFc-binding protein sequences expressed at the pediveliger stage would play a role during the crawling phase of *C. gigas*.

These proteins have adhesion properties and could also be implicated in the structure of the final adhesive. Indeed, some structures present in these sequences are common to adhesive proteins. IgGFc-binding protein sequences have been identified in adhesive footprints of starfish *Asterias rubens* [53]. The sequence EKC18867 (CGI_10010556) has seven EGF domain sites. This type of repetition has been observed in byssal plaque protein (mfp2) in mussels, which has a repetition of 11 EGF domains [54]. Sequences EKC18864, EKC18865 and EKC18868 (CGI_10010553, CGI_10010554 and CGI_10010557) have cysteine-rich regions, indicating the potential ability to establish disulfide bonds. These domains may also indicate a folding conformation in these proteins. These sequences also contain von Willebrand Factor type D (vWF-D) and EGF domains. This type of domain have been described in byssal proteins of mussels and in adhesive proteins of starfish [54,55]. The sequence EKC20685 (CGI_10005627 | Hemicentin 1) has one LDL receptor domain (low density lipoproteins) and three TSP-1 domains (type 1 thrombospondin), indicating a probable extracellular localization. The central role of a protein containing three TSP-1 domains has been reported in the byssus of *P. fucata* [56], and is probably related to the elastic properties of the distal part of the filament.

Four sequences containing an EGF-like and tenascin-related domain were selected (CGI_10010465, CGI_10000981, CGI_10025295 and CGI_10013281). Tenascins are generally extracellular, glycosylated, and have elastic properties [57]. Tenascin R may play a role in the development of the nervous system [58]. Tenascin X is a protein with essential architectural functions, implicated in the structural properties of many tissues by binding to other constitutive proteins [59]. This extracellular protein is particularly involved in cell-matrix adhesion.

Among the specific sequences of the pediveliger stage of *C. gigas*, four have a collagen triple helix domain: CGI_10010827, CGI_10010374, CGI_10010375 and CGI_10011175. Collagen is a structural protein forming fibrous structures. In marine bioadhesives, a component of the byssal filament secreted by mussels [60,61]. Sequences EKC18813 (CGI_10010827), EKC27350 (CGI_10010374) and EKC27351 (CGI_10010375) have high glycine contents (25.7%, 27.2%, and 23.3% respectively), close to the levels observed in the byssus of *P. fucata*, *M. californianus* [60] and *M. edulis* [62]. These sequences also have high proline contents, close to 11%. Glycine and proline are essential amino acids for the establishment of the triple collagen helix for fiber formation [63]. The sequence EKC27706 (CGI_10011175) has a von Willebrand Factor type A domain (vWF-A), which is a glycoprotein-binding site. The vWF-A domain is also involved in collagen binding, as previously described in mussel byssus [56,64]. According to the Phyre2 program [65], this sequence has 25% identity (99.4% confidence) with the proximal thread matrix protein (ptmp-1 | AAL17974.1) from *Mytilus galloprovincialis* [64].

The filamentous structure of the adhesive of *C. gigas* pediveliger larvae fits with a collagen-rich composition, and the cohesion of the adhesive could result from the presence of von Willebrand Factor type domains.

### 3.4. Calcifying Sequences

Redox enzymes such as tyrosinases (CGI_10009044 | EKC19270, CGI_10014286 | EKC25254 and CGI_10006802 | EKC29117), peroxidase (CGI_10016593 | EKC32997), and the two carbonic anhydrases (CGI_10010889 | EKC18733 and CGI_10011324 | EKC32754) could be related to bio-calcification. Indeed, these enzymes have been classically reported in shell synthesis processes in molluscks and could coincide with the beginning of calcite layer production, which occurs after metamorphosis, at the spat stage. These sequences are poorly expressed at the spat stage, however, compared with the pediveliger stage, although no quantitative relationship has yet been established between the RPKM value observed for any mRNA of sequences in the dataset and the abundance of the corresponding protein. Nevertheless, these sequences are also poorly expressed at the adult stage, particularly in the mantle, the organ responsible for shell synthesis. In contrast, a recent study on *C. virginica* showed the succession of two adhesion strategies during the transition of pediveliger larvae to spat [9]: the secretion of the organic adhesive by the pediveliger larva is followed by the secretion of an adhesive containing a larger inorganic fraction, allowing adherence to the substrate of the growing shell. It also appears that adhesive proteins in molluscks have similar molecular domains and functions to so-called calcifying proteins. Thus, the detection of tyrosinase, DOPA chemistry-related proteins, polysaccharide-binding domains, vWF domains and EGF domains are often common to adhesion proteins [66], including those involved in bivalve mollusck byssus [27,54] and calcification [46].

Calcium-binding sequences were selected: three protocadherin Fat 4-like proteins (CGI_10008331, CGI_10018326 and CGI_10022907) and a putative calmodulin (CGI_10006247) containing multiple EF-hand domains. An EF-hand domain consists of two alpha helix forming a loop by interaction with a Ca^2+^ ion. A sequence annotated as hypothetical protein (CGI_10025191) also presents two EF-hand domains. Extracellular proteins containing an EF-hand domain were reported in the calcification process in pearl oysters [67]. Calmodulin is a ubiquitous protein, involved in calcium metabolism. In oysters, this protein plays an important role in calcification [68]. Protocadherin Fat 4 proteins contain multiple cadherin domains, which could be involved in cell binding [69]. Many protocadherin Fat 4 proteins have been described as involved in the development of the nervous system in cephalopods [70]. In contrast, proteins containing a Ca^2+^-binding site play a role in many cellular processes (homeostasis maintenance, muscle contraction, cell differentiation, cell adhesion, immunity, signal transmission) [71].

To date, no proteins containing EF-hand domains have been reported in bioadhesives composition. However, calmodulins were detected in the adhesive organs of urchins and could play a role in exocytosis of adhesive [72]. The role of these sequences, specifically expressed at the pediveliger stage in *C. gigas*, remains unknown. It could coincide with the morphogenesis of the foot and the development of the nervous system, but also with metamorphosis or with the secretion of the shell and its adhesive matrix after metamorphosis [9].

Six sequences (CGI_10010615, CGI_10006917, CGI_10006919, CGI_10006920, CGI_10006921, and CGI_10006922) coding for proteins with a C-type lectin domain (EKC18891, EKC42164, EKC42165, EKC42166, EKC42167 and EKC42168) were selected. A C-type lectin domain is a calcium-dependent polysaccharide-binding domain [73]. These six sequences are also related to perlucin. Perlucins are proteins involved in calcification, implicated in the nucleation of calcium carbonate crystals [74]. However, the expression profile of these sequences raises doubts about their true function in *C. gigas* larvae. The sequences CGI_10010615 (EKC18891), CGI_10006917 (EKC42164) and CGI_10006920 (EKC42166) were annotated as aggrecan core protein, asialoglycoprotein receptor 2 and C-type mannose receptor 2, respectively. In vertebrates, aggrecan core protein is a constitutive protein of cartilage [75]. Perlucin-like sequences were identified in the foot of *Chlamys farreri* [12]. It is possible that these sequences are directly involved in the composition of the adhesive. Indeed, C-type lectin domains are present in many bioadhesives [47,55,76,77,78]. In addition, the sequences EKC18891, EKC42164, EKC42165, EKC42165, EKC42166, EKC42167, and EKC42168 have 46%, 48%, 44%, 47%, 44%, and 48% homology (*E*-values: 1 ×10^−47^, 2 × 10^−47^, 7 × 10^−41^, 7 × 10^−41^, 4 × 10^−45^, 2 × 10^−39^ and 3 × 10^−48^), respectively, with the foot protein 1 (AIWO4139) from *Atrina pectinata* [47]. The multiple alignment of the sequences indicates that the homology comes from the C-type lectin domain (Figure 1).

The shell of mollusck larvae is composed of a polymorphic inorganic matrix of CaCO_3_ and an organic matrix, the periostracum. The link between the adhesive and the shell could involve bonds with the periostracum. The periostracum is composed of glycoproteins and polysaccharides such as chitin [79]. It is highly likely that the C-type lectin domain present in perlucin-like sequences allows a link between the periostracum and the adhesive.

### 3.5. Hypothetical Sequences

Twenty-one sequences were annotated as hypothetical proteins, fifteen of which had extracellular localization according to DeepLoc 1.0 (CGI_10010208, CGI_10004853, CGI_10002578, CGI_10001746, CGI_10013386, CGI_10005578, CGI_10013385, CGI_10003237, CGI_10025142, CGI_10009961, CGI_10012470, CGI_10016093, CGI_10016094, CGI_10026725, and CGI_10008429). The sequence CGI_10022908 had a transmembrane domain and a predicted localization in the endoplasmic reticulum. When further information is available about these genes, it will be possible to suggest the roles they could play in adhesion.

## 4. Hypothetical Model of Molecular Interactions within *C. gigas* Adhesive

In silico analysis of transcriptomic data on the *C. gigas* oyster [24] made it possible to select transcripts that were over-expressed at the pediveliger stage. After analysis of the conserved domains and repeat sequences of the 59 selected transcripts, it appeared that the majority probably had an extracellular localization and potential roles in adhesion. These results must be treated with caution, however, based on the analysis of the transcriptome. They do not validate the presence of the proteins encoded by the identified genes. Hypothetical involvement in adhesion of some protein domains identified in genes specifically expressed in the pediveliger stage of *C. gigas* is shown in Figure 2.

Thus, the presence of structural proteins such as collagen and proteins rich in vWF domains is in accordance with the fibrous structure of the adhesive secreted by *C. gigas* pediveliger larvae. These proteins can be proposed as the structural components of the fibers and central matrix of the adhesive. In addition, the presence of vWF-A domains in an extracellular collagen sequence and vWF-D domains within an IgGFc-binding protein may indicate that these domains could be involved in the adhesive structure by mediation of protein aggregation.

The adhesive of *C. gigas* has the particularity of having its upper part in contact with the shell of the left larval valve. This shell is composed of calcium carbonate and an organic matrix, the periostracum. Since the periostracum is composed of glycosylated proteins and polysaccharides, it is likely that perlucin-like proteins containing a C-type lectin domain could mediate linkage between the adhesive and the periostracum.

The secreted adhesive needs to be resistant to the environment and not degraded until the shell of the spat has reached a sufficient size for the oyster to remain attached to the substrate. Sequences selected in this study and annotated with protease and peroxidase inhibitory domains could have a role in protecting the adhesive against bacterial degradation. In addition, the distribution of *C. gigas* on the foreshore means that the adhesive will potentially be exposed to UV radiation, desiccation, fresh water and high temperatures, which could be stress factors for this material.

The chemical bonds involved in the structure of the adhesive, and in linkage to the substrate, could be of a diversity of types. The cysteine-rich domains observed in some sequences (EKC18864 | CGI_10010553, EKC18865 | CGI_10010554, and EKC18868 | CGI_10010557) may be involved in the structural conformation of these proteins, but also in the establishment of disulfide bonds. Disufide bonds can also be established between other proteins containing cysteines in their sequences. The selection of sequences coding for enzymes involved in redox mechanisms (tyrosinases, peroxidase) could potentially indicate the biosynthesis of DOPA and these derivatives. This makes it possible to consider adhesion mechanisms in pediveliger larvae similar to those described in the byssus of adult bivalve molluscks. Peroxidases and tyrosinases can thus allow the formation of dityrosine bonds, and DOPA or DOPA-quinone groups, respectively. DOPA can mediate linking between adhesive and substrate, but also between the proteins in the adhesive, by covalent bonds or ionic interactions. Ionic interactions could also occur within the adhesive, particularly due to the presence of sequences with domains having a calcium affinity (cadherin, EGF, EF-hand). It is possible that the difference in structure observed between the inner and outer zone of the adhesive in *C. gigas* [4] may result from the presence in the outer zone of proteins or chemical groups allowing the establishment of a greater number of covalent bonds or ionic interactions, resulting in a tightening of the adhesive mesh.

## 5. Materials and Methods

Table S14 “Transcriptomic representation of genes (RPKM) at different developmental stages and in different adult organs” was downloaded from the *C. gigas* genome publication on the Nature website [24]. Stages were defined by the number of days after fertilization of the larval cohort. Thus, P1 and P2 were defined as the pediveliger stage at 18 days post-fertilization (precisely sampled at 18 days and 45 min and at 18 days, 4h and 35 min after fertilization). In order to identify genes potentially involved in adhesion, sequences with “strong” RPKM at stages P1 and P2 relative to the other stages were selected according to the following thresholds:RPKM of stages E to U6, stages S and J, and adult organs less than 20% that of P1 or P2.RPKM of stages LU1, LU2 below 70% that of P1 or P2.

LU1 and LU2 correspond to “later umbo larvae 1” and “later umbo larva 2” at 14 and 15 days post-fertilization, respectively. The RPKM selection threshold of these two stages was higher than for other larval stages according to the potential heterogeneity of the larval cohort and potential RNA synthesis before settlement at the pediveliger stage.

The protein sequences corresponding to the selected genomic sequences were then searched for on the NCBI database. Functional annotation of protein sequences was performed with BLAST2GO 4.0.7 software. A BLASTP search was performed with BLASTP2.7.1 + (NCBI) from BLAST2GO in the non-redundant protein database (nr) on 23 August 2018. Conserved domains and repetitive sequences were predicted with InterPro [25], and subcellular localization was predicted with DeepLoc 1.0 program [26]. Sequences of interest were then also submitted to Phyre2 to search for protein similarities based on protein structure prediction [65].

## 6. Conclusions

Our in silico analysis successfully identified genes specifically expressed at the pediveliger stage in *C. gigas*. The majority of these genes were annotated for proteins containing conserved domains potentially involved in adhesion of *C. gigas*. Hypotheses formulated during this analysis advanced our understanding of larval adhesion in *C. gigas*. The set of 59 selected transcripts found by this method are interesting candidates that could be functionally explored by localization approaches (RNA in situ hybridization, antibody hybridization) and phenotyping approaches (interfering RNA, CRISPR). Thus, the localization of these transcripts and/or of the corresponding proteins within the pediveliger larvae and/or adhesive imprints is needed in order to validate their respective involvement in adhesion. The localization of these proteins would also make it possible to define molecular interactions and bonds involved in the structure of this adhesive. However, our results are based on gene expression and are thus not a perfect reflection of the real protein composition of the adhesive. Indeed, despite the precautions taken in the sequence selection protocol (time window tightened around the pediveliger stage), regulatory mechanisms between the mRNA and its translation into protein are multiple. Larval adhesives are difficult to describe and characterize because of the small amount of material and the resistance of these matrices. Our results represent significant progress about oyster larval adhesive, and open new possibilities for further analysis on this bioadhesive.

## Figures and Tables

**Figure 1 ijms-20-00197-f001:**
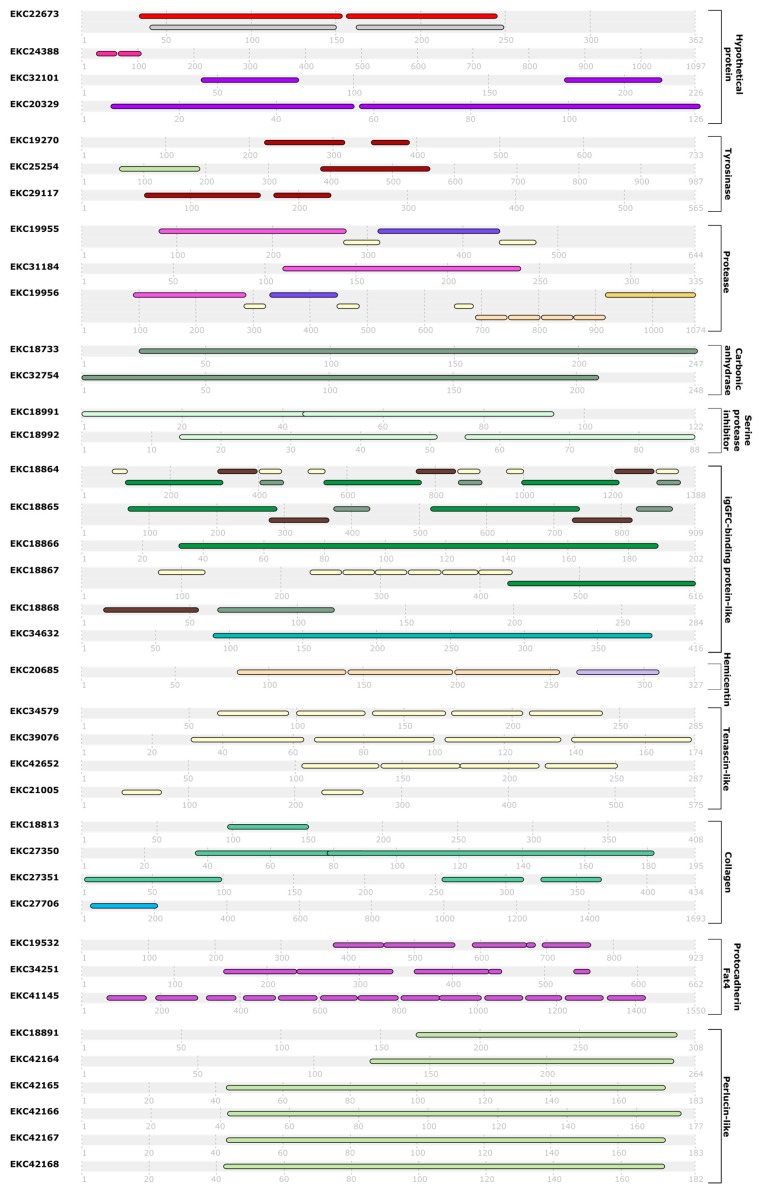
Conserved domains and repeated sequences predicted by the InterPro program (Finn et al., 2016) [25] among 38 sequences specifically expressed at the pediveliger stage in *Crassostrea gigas*, based on transcriptomic data published by Zhang et al. (2012) [24].

**Figure 2 ijms-20-00197-f002:**
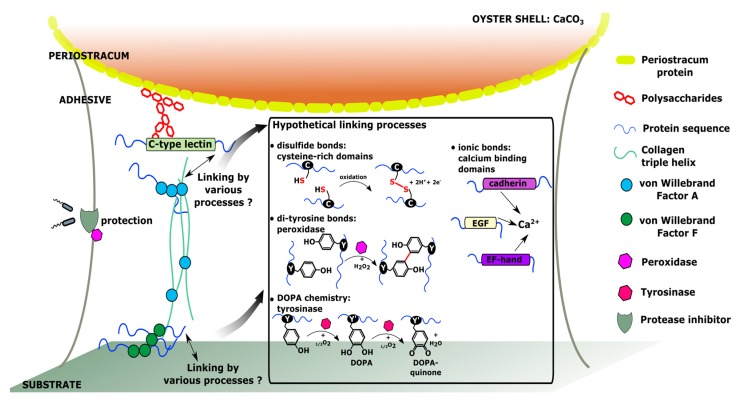
Schematic representation of the hypothetical molecular interactions involved in the adhesion of *C. gigas* pediveliger larvae, based on the selection of genes specifically expressed at the pediveliger stage.

**Table 1 ijms-20-00197-t001:** Genes specifically expressed at the pediveliger stage of *Crassostrea gigas* according to the selection of RPKM from transcriptomic data published by Zhang et al. (2012) [24].

Group	Ensembl Gene ID	Protein ID	Name	Cell. Loc.
Hypothetical protein	CGI_10014580	EKC18206	Hypothetical protein CGI_10014580	Mit 0.58
	CGI_10010208	EKC18972	Hypothetical protein CGI_10010208	Ext 1
	CGI_10004853	EKC21005	Hypothetical protein CGI_10004853	Ext 0.42
	CGI_10002578	EKC22248	Hypothetical protein CGI_10002578	Ext 0.89
	CGI_10001746	EKC22673	Hypothetical protein CGI_10001746	Ext 0.90
	CGI_10013335	EKC23310	Hypothetical protein CGI_10013335	Nuc 0.50
	CGI_10013386	EKC24388	Hypothetical protein CGI_10013386	Ext 0.41
	CGI_10005578	EKC25384	Hypothetical protein CGI_10005578	Ext 0.43
	CGI_10013385	EKC24387	Hypothetical protein CGI_10013385	Ext 0.60
	CGI_10003237	EKC27225	Hypothetical protein CGI_10003237	Ext 0.99
	CGI_10025142	EKC28625	Hypothetical protein CGI_10025142	Ext 0.91
	CGI_10009961	EKC31321	Hypothetical protein CGI_10009961	Ext 0.99
	CGI_10025191	EKC32101	Hypothetical protein CGI_10025191	Cyt 0.38
	CGI_10012470	EKC33059	Hypothetical protein CGI_10012470	Ext 0.99
	CGI_10016093	EKC35263	Hypothetical protein CGI_10016093	Ext 0.52
	CGI_10016094	EKC35264	Hypothetical protein CGI_10016094	Ext 0.52
	CGI_10027526	EKC35968	Hypothetical protein CGI_10027526	ER 0.40
	CGI_10026725	EKC38958	Hypothetical protein CGI_10026725	Ext 0.62
	CGI_10022908	EKC41146	Hypothetical protein CGI_10022908	ER 0.25
	CGI_10008429	EKC41249	Hypothetical protein CGI_10008429	Ext 0.90
	CGI_10013282	EKC42653	Hypothetical protein CGI_10013282	Nuc 0.59
Enzyme	CGI_10009044	EKC19270	Putative tyrosinase-like protein tyr 1	Mem 0.82
	CGI_10014286	EKC25254	Putative tyrosinase-like protein tyr-3	Mem 0.99
	CGI_10006802	EKC29117	Tyrosinase-like protein 1	Ext 0.64
	CGI_10016593	EKC32997	Peroxidase-like protein	Ext 0.49
	CGI_10010889	EKC32754	Carbonic anhydrase 2	Cyt 0.41
	CGI_10011324	EKC18733	Carbonic anhydrase 7	Ext 0.65
	CGI_10003099	EKC28981	Cell surface hyaluronidase-like	Plast 0.39
	CGI_10003100	EKC28982	Cell migration-inducing and hyaluronan-binding protein-like	Cyt 0.29
	CGI_10007190	EKC19955	Metalloendopeptidase	Ext 0.39
	CGI_10007191	EKC19956	Metalloendopeptidase	Ext 0.51
	CGI_10020760	EKC31184	Zinc metalloproteinase nas-15	Ext 0.70
Protease inhibitor	CGI_10010154	EKC18991	Serine protease inhibitor dipetalogastin-like	Ext 0.55
	CGI_10010155	EKC18992	Serine protease inhibitor dipetalogastin-like	Ext 1
Structural protein	CGI_10005627	EKC20685	Hemicentin-1	Ext 0.72
	CGI_10010553	EKC18864	IgGFc-binding protein (zonadhesin-like)	Ext 0.64
	CGI_10010554	EKC18865	IgGFc-binding protein (zonadhesin-like)	Ext 0.72
	CGI_10010555	EKC18866	IgGFc-binding protein (zonadhesin-like)	Ext 0.93
	CGI_10010556	EKC18867	IgGFc-binding protein (zonadhesin-like)	Ext 0.60
	CGI_10010557	EKC18868	IgGFc-binding protein (zonadhesin-like)	Ext 0.91
	CGI_10023170	EKC34632	IgGFc-binding protein	Ext 0.52
	CGI_10010465	EKC34579	Tenascin-X	Ext 0.80
	CGI_10000981	EKC39076	Tenascin-R	Ext 0.84
	CGI_10025295	EKC40994	Multiple EGF-like domains 10	Ext 0.42
	CGI_10013281	EKC42652	Tenascin-R	Ext 0.99
	CGI_10010827	EKC18813	Collagen-like protein 7	Ext 0.89
	CGI_10010374	EKC27350	Collagen-like protein 7	Mem 0.56
	CGI_10010375	EKC27351	Collagen-like protein 7	Mem 0.72
	CGI_10011175	EKC27706	Collagen alpha-5(VI) chain	Ext 0.55
Calcification-related protein and calcium-binding protein	CGI_10010615	EKC18891	Aggrecan core protein	Ext 0.79
	CGI_10006917	EKC42164	Asialoglycoprotein receptor 2	Ext 0.99
	CGI_10006919	EKC42165	Perlucin-like protein	Ext 0.95
	CGI_10006920	EKC42166	C-type mannose receptor 2	Ext 0.90
	CGI_10006921	EKC42167	Perlucin-like protein	Ext 0.99
	CGI_10006922	EKC42168	Perlucin-like protein	Ext 0.99
	CGI_10008331	EKC19532	Protocadherin Fat 4-like	Lys 0.36
	CGI_10018326	EKC34251	Protocadherin Fat 4	Cyt 0.72
	CGI_10022907	EKC41145	Protocadherin Fat 4-like	Cyt 0.44
	CGI_10006247	EKC20329	Putative calmodulin	Cyt 0.48

Cell. Loc. indicates the subcellular localization prediction of the corresponding protein by DeepLoc 1.0. Ext: extracellular, Cyt: cytoplasm, Lys: lysosome, Mit: mitochondria, Pla: plastid, Mem: cell membrane, ER: endoplasmic reticulum, Nuc: nucleus.

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
