# Peer review of "In Silico Analysis of Pacific Oyster (Crassostrea gigas) Transcriptome over Developmental Stages Reveals Candidate Genes for Larval Settlement"

_ijms, 2019, doi:10.3390/ijms20010197_

Round 1
Reviewer 1 Report
Foulon et al. present an entirely ‘in silico’ analysis of up-regulated transcripts in the pediveliger (attachment) stage of oysters, with the intention of identifying proteins involved in the process of adhesion.
The source material for this analysis is the original publication of the oyster genome. I emphasis this, because there is an important distinction between post hoc analyses like this and experimental research that generates empirical data. The reader must be aware that the findings of this work are entirely hypothetical and have no experimental evidence presented to support them. The proposals for possible molecular adhesion processes highlighted in the article are interesting and well-founded in the literature. It is also a good ‘starting point’ for future experimental studies into the oyster adhesion system. But, given that the data were already published and that the mechanisms proposed are those commonly discussed in the literature, it is difficult to find much novelty in this study.
Taken for what it is, the article is nevertheless of good quality. The methods and interpretation seem fine, and it is well presented. I have listed some minor corrections below.
I have two main areas of doubt. First is the entirely theoretical approach taken by the authors, discussed above. Personally I do not like this and I do not think it contributes a great deal, but if it is acceptable to the journal then I would support publication. The second point is that the findings are based entirely on mRNA sequencing data, and extrapolated to the protein/interactions level. This is very risky. The authors acknowledge that there are temporal limitations on the validity of this correlation (line 77), but I feel they have significantly understated the case. The relationship between mRNA and proteins present at any given time can be very weak indeed. Since there is no experimental validation here (e.g. peptide mass fingerprinting, MS/MS), it is possible, for example, that the transcripts described in the paper have nothing at all to do with adhesion and that they are in fact involved in future growth processes. Can the authors confirm somehow that this is not the case? Do they have any proteomics data that could validate the transcript IDs?
In summary, the paper is well written and provides ‘food for thought’, but in my opinion it does not actually provide any evidence or novel insight.
Minor corrections:
Abstract:
Line 17 - databases
Line 19 - genes
Line 21 - Some related proteins contain conserved…
Line 22 - Remove comma after ‘model’
Introduction:
Line 31 - wet conditions
Line 31 - , in particular for biomedical applications
Line 32 - composition, especially proteins, from…is sometimes difficult
Line 39 - remove italic font ‘of’
Line 42 - larvae remain unknown
Line 80 - In our study…
Line 86 - sequence
Line 87 - organs
Line 88 - I am not familiar with the time format used here (18.03 and 18.19). Some explanation?
Line 89 - The objective was (past tense)
Line 92 - involved in their secretion
Results:
Table 1 caption - italics for Crassostrea gigas
Figure 1 caption - italics for Crassostrea gigas
Line 130 - remove ‘reactions’
Line 156 - remove ‘interestingly’
Line 161 - The roles of these enzymes remain uncertain
Line 163 - signal peptide
Line 164 - remove ‘an’
Line 166 - peroxidases has
Line 173 - Avoid one sentence paragraphs
Line 177 - These enzymes are
Line 178 - and are important regulators of…
Line 180 - active domains were..
Line 189 - 192 - I find this passage tenuous and suggest removing it.
Line 231 - pathogens
Line 232 - binding proteins
Line 235 - contents
Line 237 - Avoid one sentence paragraphs
Line 261 - remove ‘the’
Line 269 - …site .vWF-A domains are…
Line 273 - Avoid one sentence paragraphs
Line 306 - domains
Line 309 - remove ‘preparation’
Line 312 - proteins
Line 313 - domain
Line 330 - It is likely that…
Line 332 - italicise C. gigas
Line 336 - …localisation and potential roles in adhesion.
Figure 2 - bottom left - substrate
Figure 2 legend - italicise C. gigas
Line 345 - components
Line 354 - needs
Line 355 - has reached sufficient size for the oyster to remain attached to the substrate.
Line 358 - foreshore subjects the adhesive to UV…
Line 361 - could be diverse
Line 407 - sequences of interest
Author Response
Response to reviewer 1
Dear reviewer,
We would like to thank you for your constructive comments that helped improving our manuscript. This letter presents point-by-point response to your comments. Each comment has been carefully addressed.
Yours faithfully,
Valentin Foulon on behalf of all co-authors.
Comments and suggestion for Authors
Foulon et al. present an entirely ‘in silico’ analysis of up-regulated transcripts in the pediveliger (attachment) stage of oysters, with the intention of identifying proteins involved in the process of adhesion.
The source material for this analysis is the original publication of the oyster genome. I emphasis this, because there is an important distinction between post hoc analyses like this and experimental research that generates empirical data. The reader must be aware that the findings of this work are entirely hypothetical and have no experimental evidence presented to support them. The proposals for possible molecular adhesion processes highlighted in the article are interesting and well-founded in the literature. It is also a good ‘starting point’ for future experimental studies into the oyster adhesion system. But, given that the data were already published and that the mechanisms proposed are those commonly discussed in the literature, it is difficult to find much novelty in this study.
Taken for what it is, the article is nevertheless of good quality. The methods and interpretation seem fine, and it is well presented. I have listed some minor corrections below.
I have two main areas of doubt. First is the entirely theoretical approach taken by the authors, discussed above. Personally I do not like this and I do not think it contributes a great deal, but if it is acceptable to the journal then I would support publication.
For this first point, we are agree on the theoretical aspect of our work. However, our approach highlighted for the first time some candidates genes potentially implicated in larval settlement of C. gigas larvae. It also indicate that post hoc analysis could be a useful tool as ‘starting point’ for larval bioadhesive research. Of course, the data were already published with oyster genome. Of course, literature concerning marine bioadhesive is available. Our study make the link between them and valorize the public oyster transcriptome data.
We think that our results could contribute to the research advancement of larval bioadhesive that are poorly documented because of the difficulty to study it.
The second point is that the findings are based entirely on mRNA sequencing data, and extrapolated to the protein/interactions level. This is very risky. The authors acknowledge that there are temporal limitations on the validity of this correlation (line 77), but I feel they have significantly understated the case. The relationship between mRNA and proteins present at any given time can be very weak indeed. Since there is no experimental validation here (e.g. peptide mass fingerprinting, MS/MS), it is possible, for example, that the transcripts described in the paper have nothing at all to do with adhesion and that they are in fact involved in future growth processes.
This second point was highlighted in the revised manuscript,
in the discussion. Line 178: “. It is therefore very likely that these two mRNA are expressed at the pediveliger stage in order to prepare the rapid calcification observed after metamorphosis.”
In the conclusion : line 398 “Indeed, despite the precautions taken in the sequence selection protocol (time window tightened around the pediveliger stage), regulatory mechanisms between the mRNA and its translation into protein are multiple.”
Can the authors confirm somehow that this is not the case? Do they have any proteomics data that could validate the transcript IDs?
We can not confirm our hypothesis at this point. Proteomic identification of oyster larval adhesive is highly tricky. In C. gigas larvae, around 200pL of adhesive is produced during settlement. However, we are currently working on proteomic analysis. First results seems to validate some results of this in silico analysis. But our data are not ready for publication, and some point are in progress. We think that the publication of this paper is important for the community even if it base on hypothesis.
In summary, the paper is well written and provides ‘food for thought’, but in my opinion it does not actually provide any evidence or novel insight.
All minor corrections were addressed in the revised manuscript.
Reviewer 2 Report
The work presented by Foulon and collaborators gives us an in silico analysis of transcripts from the pacific oyster Crassostrea gigas. Transcriptomic data was obtained from previous research by Zhang G. and collaborators (DOI: 10.1038/nature11413) and was analyzed to look for candidate genes involved in adhesion. The authors retrieve 59 candidate sequences that are over-expressed in the larval stage of the oyster. Then, using bioinformatic tools, the domains of the sequences are inferred and compared to similar domains found in sequences of other marine animals that lay adhesives. While the discussion section of the manuscript is strong, there are many questions that arise from claims made by the authors.
For example:
The rationale behind the sample and threshold selection is not clear. The materials and method section mentions “strong” RPKM values with respect to other stages. What is a strong RPKM value? Is there a gene that could be used as a baseline to define this? What RPKM value is going to allow us to see if a gene is being over expressed? A figure showing the comparison between strong and weak RPKM values is and where the candidate sequences are could be useful.
In line 159, the authors mention that CGI_10016593 had a 44.8% similarity to a protein from Mytilus coruscus. Was BLAST used to align the sequences and verify the similarity? If a BLAST alignment was performed, what parameters were used? This question comes to mind because the reference cited is from a protein sequence not an RNA sequence.
In line 171, the authors mention that the observations and discussion indicate the involvement of Dihydroxyphenyl Alanine (DOPA) biosynthesis. Is there a way that this claim could be validated? This section in particular does not have a reference to support the claim. This happens again in line 182 when carbonic anhydrase is mentioned.
In line 334, the authors claim that the in silico analysis “made it possible to select transcripts that were overwhelmingly expressed”. As mentioned before, what does this mean in terms of RPKM value? If another research group wanted to perform this analysis, what would be the workflow, parameters and thresholds that could be used to identify sequence candidates with certainty?
In line 343, the authors propose a model in which collagen and vWF domains are involved in adhesion. There was no previous discussion of this in the manuscript, so is there any evidence to support this claim? Is there a way to verify that the transcripts are being translated into proteins? Are there any references that can support this claim with transcriptomic and proteomic data?
In the conclusions, the authors mention several techniques to test the hypotheses described in the discussion. Could any of these techniques be performed for this manuscript? If not, is there a way to validate the bioinformatic work that was shown in the manuscript?
It is clear that the authors have the background and knowledge of this topic. However, even with great analysis and discussion, there are claims and assumptions that need to be validated. This is particularly true in bioinformatics where each experiment has to be supported by statistics. In its current form, I would not accept this manuscript for publication.
Author Response
Response to reviewer 2
Dear reviewer,
We would like to thank you for your constructive comments that helped improving our manuscript. This letter presents point-by-point response to your comments. Each comment has been carefully addressed.
Yours faithfully,
Valentin Foulon on behalf of all co-authors.
The work presented by Foulon and collaborators gives us an in silico analysis of transcripts from the pacific oyster Crassostrea gigas. Transcriptomic data was obtained from previous research by Zhang G. and collaborators (DOI: 10.1038/nature11413) and was analyzed to look for candidate genes involved in adhesion. The authors retrieve 59 candidate sequences that are over-expressed in the larval stage of the oyster. Then, using bioinformatic tools, the domains of the sequences are inferred and compared to similar domains found in sequences of other marine animals that lay adhesives. While the discussion section of the manuscript is strong, there are many questions that arise from claims made by the authors.
For example:
The rationale behind the sample and threshold selection is not clear. The materials and method section mentions “strong” RPKM values with respect to other stages. What is a strong RPKM value? Is there a gene that could be used as a baseline to define this? What RPKM value is going to allow us to see if a gene is being over expressed? A figure showing the comparison between strong and weak RPKM values is and where the candidate sequences are could be useful.
As described in the material and method Line 409: “In order to identify genes potentially involved in adhesion, sequences with "strong" RPKM at stages P1 and P2 relative to the other stages were selected according to the following thresholds:
● RPKM of stages E to U6, stages S and J, and adult organs less than 20% that of P1 or P2.
● RPKM of stages LU1, LU2 below 70% that of P1 or P2.”
This indicates that the RPKM value at the pediveliger stage is 5 times higher than other larval or adult stages (exception for LU1 and LU2 stages).
Material and method section was completed with the following information:
Line 414: “LU1 and LU2 correspond to “later umbo larvae 1” and “later umbo larva 2” at 14 and 15 days post-fertilisation, respectively. The RPKM selection threshold of these two stages was higher than for other larval stages according to the potential heterogeneity of the larval cohort and potential RNA synthesis before settlement at the pediveliger stage.
In line 159, the authors mention that CGI_10016593 had a 44.8% similarity to a protein from Mytilus coruscus. Was BLAST used to align the sequences and verify the similarity? If a BLAST alignment was performed, what parameters were used? This question comes to mind because the reference cited is from a protein sequence not an RNA sequence.
BLAST parameters used were add in the material and method line 420:
“A BLASTP search was performed with BLASTP2.7.1+ (NCBI) from BLAST2GO in the non-redundant protein database (nr) on 08/23/2018.”
E-value from BLASTP result was added in the text for all similarities (Lines 145, 146, 155 and 319).
In line 171, the authors mention that the observations and discussion indicate the involvement of Dihydroxyphenyl Alanine (DOPA) biosynthesis. Is there a way that this claim could be validated? This section in particular does not have a reference to support the claim. This happens again in line 182 when carbonic anhydrase is mentioned.
As indicate in the text Line 181, “The presence of tyrosinase and peroxidase coding sequences in our C. gigas pediveliger larva sequence selection could indicate a use of redox mechanisms in its adhesive with the presence of DOPA or phenol groups.“ this observation is hypothetic.
As for the role of carbonic anhydrase, no enzyme assay or proteomic were performed to validate our hypotheses. This was not the propos of this paper. Our results propose by the valorization of public transcriptome database, and solid references, a hypothetic composition of the larval adhesive of C. gigas. We think that that this paper is important for the community, and could motivate further investigation on the numerous candidates highlighted here.
In line 334, the authors claim that the in silico analysis “made it possible to select transcripts that were overwhelmingly expressed”. As mentioned before, what does this mean in terms of RPKM value? If another research group wanted to perform this analysis, what would be the workflow, parameters and thresholds that could be used to identify sequence candidates with certainty?
“to select transcripts that were overwhelmingly expressed “ indicate that is possible to select transcript that have a RPKM value relatively high compared with another. Workflow parameters were described in the material and method. If another research group wanted to perform this type of analysis, it needs to conceptualize a workflow adapted to the biological question, as it was made in our analysis. Our method was adjusted to identify sequences potentially implicated in bioadhesion. It comes from our knowledges about C. gigas larval biology, settlement process, foot morphogenesis, and adhesive secretion… And as indicate in our paper, results from this type of analysis are hypothetic, there is no certainty, but it presents many indications for further analysis.
In line 343, the authors propose a model in which collagen and vWF domains are involved in adhesion. There was no previous discussion of this in the manuscript, so is there any evidence to support this claim? Is there a way to verify that the transcripts are being translated into proteins? Are there any references that can support this claim with transcriptomic and proteomic data?
This hypothesis was based on our results and references cited in the text line 264. Collagen and vWF were already reported in other marine bioadhesive (mussel byssus [56, 64]).
In the conclusions, the authors mention several techniques to test the hypotheses described in the discussion. Could any of these techniques be performed for this manuscript? If not, is there a way to validate the bioinformatic work that was shown in the manuscript?
There is several way to validate the bioinformatics work shown in the manuscript (proteomic analysis, transcriptomic analysis on adhesive glands, ISH, immunolabeling, RNAi…) but is not the goal of this manuscript. We think that this work is sufficient for publication, as it present for the first time a possible global composition of oyster adhesive. Marine larval adhesive are poorly documented because of the difficult to study it (very few amount of material, resistance of secreted bioadhesive). Our study presents an alternative with interesting results, hypothetic, but that could be validated, complemented and discussed by further studies.
It is clear that the authors have the background and knowledge of this topic. However, even with great analysis and discussion, there are claims and assumptions that need to be validated. This is particularly true in bioinformatics where each experiment has to be supported by statistics. In its current form, I would not accept this manuscript for publication.
Reviewer 3 Report
This study identifies differentially expressed genes between life stages of the Pacific oyster by transcriptome analysis in order to provide preliminary conclusions on what genes may be involved in production of bioadhesive by the larval stage. This is an interesting study that yields important findings. However, the study design is of low originality and the data is very preliminary and lacks additional supportive evidence. This manuscript could be greatly strengthened by complimentary experiments to support the tentative conclusions drawn from the transcriptomics. This manuscript needs moderate to extensive English language editing in all sections.
Abstract:
The claim that the authors propose a model (line 22) is a little misleading, as a model implies a representative system developed to study a particular phenomenon. This phrasing could be changed to “we propose that collagen comprises the proteinaceous component of the bioadhesive” or similar.
Introduction:
Good summary of the background, rationale, and objective of the current study. Based on summaries of other articles presented in this section, the current study is not unique in its approach but may provide useful information since it is applied to a different species. Mention of proteomics used in these other studies raises the question of why the authors of the current study did not follow up their transcriptomics with proteomics or other experiments to further strengthen the conclusions.
Results:
In lines 101-102, the sentence beginning “The number of uncharacterized…” seems to contradict the previous sentence. This may be an English language issue or a typo. The caption for Table 1 could be simplified by explaining the selection criteria in the text rather than in the caption. Please rephrase that InterPro predicts domains rather than detecting them.
Discussion:
This section is well organized by subsection. Overall it provides clear discussion of the results and proposes interesting conclusions; however, these conclusions are very preliminary without additional experiments. A minor point: in both this section and the Results section, you may want to consider putting the Hypothetical Proteins results/discussion last rather than first, since little is known about these and it is better to begin stronger.
Materials and Methods:
It is unclear what separates P1 and P2 stages, since they are stated both to be 18 d after fertilization. Please clarify this. Please also define the other life stages more thoroughly (this may alternatively be done in the Introduction) and make sure to define all abbreviations. Specify the purpose of submitting the sequences of interest to the Phyre2 program. Change line 406 to “Predicted conserved domains and repetitive sequences…”
Author Response
Response to reviewer 3
Dear reviewer,
We would like to thank you for your constructive comments that helped improving our manuscript. This letter presents point-by-point response to your comments. Each comment has been carefully addressed.
Yours faithfully,
Valentin Foulon on behalf of all co-authors.
Comments and suggestion for Authors
This study identifies differentially expressed genes between life stages of the Pacific oyster by transcriptome analysis in order to provide preliminary conclusions on what genes may be involved in production of bioadhesive by the larval stage. This is an interesting study that yields important findings. However, the study design is of low originality and the data is very preliminary and lacks additional supportive evidence. This manuscript could be greatly strengthened by complimentary experiments to support the tentative conclusions drawn from the transcriptomics. This manuscript needs moderate to extensive English language editing in all sections.
We are agree on the theoretical aspect of our work. This paper propose some candidate genes that could be investigated in further analysis concerning oyster larval bioadhesive studies. This point is highlighted in the title of the manuscript with the term ‘in silico analysis’. It present a valorization of public transcriptome of C. gigas that could motivate some further analysis in oyster larval bioadhesive.
Abstract:
The claim that the authors propose a model (line 22) is a little misleading, as a model implies a representative system developed to study a particular phenomenon. This phrasing could be changed to “we propose that collagen comprises the proteinaceous component of the bioadhesive” or similar.
The sentence was changed in the revised manuscript Line 21: “We propose a hypothetic composition of C. gigas bioadhesive in which the protein constituent is probably composed of collagen (…).“
Introduction:
Good summary of the background, rationale, and objective of the current study. Based on summaries of other articles presented in this section, the current study is not unique in its approach but may provide useful information since it is applied to a different species. Mention of proteomics used in these other studies raises the question of why the authors of the current study did not follow up their transcriptomics with proteomics or other experiments to further strengthen the conclusions.
Concerning the use of proteomic in other studies, they were made on bigger sample or bigger organisms than oyster larvae. The main challenge in oyster larval bioadhesive characterization is the few amount of available material (around 200pL of adhesive by larvae) and the resistance to protein extraction of the secreted adhesive. However, we work on the optimization of our proteomic protocol to strengthen our hypothesis based on transcriptome analysis. Even if our conclusion are hypothetic, we think that the publication of these results could help and motivate further investigations in larval bioadhesive research.
Results:
In lines 101-102, the sentence beginning “The number of uncharacterized…” seems to contradict the previous sentence. This may be an English language issue or a typo. The caption for Table 1 could be simplified by explaining the selection criteria in the text rather than in the caption. Please rephrase that InterPro predicts domains rather than detecting them.
Sentence line 107 was changed with English language correction.
Selection criteria were placed in the text and deleted from caption of Table 1.
Your relevant remark concerning InterPro prediction was addressed in the revised manuscript.
Discussion:
This section is well organized by subsection. Overall it provides clear discussion of the results and proposes interesting conclusions; however, these conclusions are very preliminary without additional experiments. A minor point: in both this section and the Results section, you may want to consider putting the Hypothetical Proteins results/discussion last rather than first, since little is known about these and it is better to begin stronger.
This remark was addressed in the revised manuscript. Hypothetical proteins section was put last in Results and Discussion.
Materials and Methods:
It is unclear what separates P1 and P2 stages, since they are stated both to be 18 d after fertilization. Please clarify this. Please also define the other life stages more thoroughly (this may alternatively be done in the Introduction) and make sure to define all abbreviations. Specify the purpose of submitting the sequences of interest to the Phyre2 program. Change line 406 to “Predicted conserved domains and repetitive sequences…”
Clarification concerning P1 and P2 stages was added in Introduction: Line 88 “The pediveliger stages were named P1 and P2 and correspond to larvae of 18 days old larvae (precisely sampled at 18 days and 45 minutes and at 18 days, 4h and 35 minutes after fertilisation).”
The purpose of submitting the sequences to the Phyre2 program was explained Line 423: “Sequences of interest were then also submitted to Phyre2 to search for protein similarities based on protein structure prediction [65].”
Line 406 was changed in Line 422: “Conserved domains and repetitive sequences were predicted with InterPro [25], and subcellular localisation was predicted with DeepLoc 1.0 program [26]”